# The Assessment of the Socioemotional Disorder in Neurodegenerative Diseases with the Revised Self-Monitoring Scale (RSMS)

**DOI:** 10.3390/jcm11247375

**Published:** 2022-12-12

**Authors:** Tatiana Dimitriou, Theodore Parthimos, Vasiliki Kamtsadeli, Niki Tsinia, Maria Hatzopoulou, Evi Lykou, Lina Chatziantoniou, Olga Papatriantafyllou, Chara Tzavara, Panagiotis Zikos, Sokratis Papageorgiou, Bruce Miller, Kate Rankin, John Papatriantafyllou

**Affiliations:** 1Department of Neurology, Aristotle University of Thessaloniki, Greece & Third Age Center IASIS, 54636 Athens, Greece; 2Division of Psychology, Faculty of Life and Health Sciences, De Montfort University, Leicester LE1 9BH, UK; 3Third Age Center IASIS, 54636 Athens, Greece; 41st Department of Psychiatry, University of Athens Medical School, Eginition Hospital, Greece & Third Age Center IASIS, 11528 Athens, Greece; 5Neurologist, 251 Hellenic Airforce Hospital, 11525 Athens, Greece; 6Department of Neurology, University of Athens Medical School, Eginition Hospital, 11528 Athens, Greece; 7Memory and Aging Center, Department of Neurology, Global Brain Health Institute, University of California, San Francisco, CA 94158, USA; 8Memory and Aging Center, Department of Neuropsychologist, University of California, San Francisco, CA 94158, USA

**Keywords:** RSMS, neurodegenerative dementias, socioemotional cognition, theory of mind, differential diagnosis

## Abstract

Background: Social cognition helps people to understand their own and others’ behavior and to modulate the way of thinking and acting in different social situations. Rapid and accurate diagnoses of neurodegenerative diseases are essential, as social cognition is affected by these diseases. The Revised Self-Monitoring Scale (RSMS) is a scale that detects social–emotional cognition deficits. Aim: The aim of the current study is to examine how socioemotional parameters are affected by neurodegenerative diseases and whether the RSMS can discern these disorders based on the socioemotional parameters in the Greek population. Methods/Design: A total of 331 dementia subjects were included. Mini Mental State Examination (MMSE) and Addenbrooke’s Cognitive Examination (Revised, ACE-R) measurements were used in order to assess the cognitive deficits. The Neuropsychiatric Inventory (NPI) was used for the evaluation of the neuropsychiatric symptoms. The RSMS and its two subscales was used in order to detect the socioemotional deficits. Results: The RSMS and its two subscales (RSMS_EX and RSMS_SP) can effectively detect neurodegenerative diseases. The RSMS can detect bvFTD in Alzheimer’s Disease (AD), AD in a healthy cohort, behavioral variant Frontotemporal Dementia (bvFTD) in a healthy cohort, bvFTD in Parkinson’s Disease (PD) and Frontotemporal Semantic Dementia (FTD/SD) in a healthy cohort. It is a useful tool in order to detect frontotemporal dementias. RSMS correlated negatively with the NPI questionnaire total and the subcategories of apathy, disinhibition and eating disorders. The RSMS results are associated with the ACE-R score (specifically verbal fluency). Conclusions: The RSMS is a helpful tool in order to identify socioemotional deficits in neurodegenerative dementias. It is also a useful scale that can discern bvFTD and svPPA in AD patients. A worse RSMS score correlates with a worse ACE-R and NPI. It seems to be a useful scale that can reliably measure social behavior in non-reversible neurodegenerative disorders, such as AD, FTD (bvFTD, svPPA), PDD and PD. The results also apply to the Greek population.

## 1. Introduction

Social cognition is the cognitive ability of how we create representations of other people and how these representations guide our actions, social perceptions and judgments [1]. It refers to a complex set of abilities. Social cognition develops in infancy and it is now recognized as a core domain of functioning that needs attention across neurological conditions [2]. Theory of Mind (ToM) is a theory that explains the ability to represent others’ mental states [3]. It refers to humans’ ability to represent the mental state of others, including their desires, beliefs and intentions [3]. According to previous studies, the dorsomedial (pregenual) prefrontal cortex, the anterior temporal lobe and the right temporoparietal junction play a major role in ToM [4,5].

Alzheimer’s Disease (AD) is the most common type of dementia and accounts for up to 60% of dementia cases. However, social cognition in AD patients has been studied less widely than in Frontotemporal Dementia (FTD) [6]. AD patients show impairments in social behavior, ToM and emotion perception [6]. These symptoms increase caregivers’ burdens because they are associated with a reduced quality of life [7]. Deficits in social cognition may be present in the Mild Cognitive Impairment stage (MCI), referred to as amnestic MCI (aMCI) [8]. Social cognition deficits in AD are related to the progression of the disease and lead to global cognition deterioration [9]. Social cognition is mainly associated with declines in executive functions [10].

Frontotemporal Dementia (FTD) can be categorized in two categories: (a) behavioral variant Frontotemporal Dementia (bvFTD) (which involves changes in personality, judgement, cognition and behavior) and (b) Primary Progressive Aphasia (PPA), which involves declines in the ability to communicate. PPA has three subcategories [11]: (i) semantic PPA (FTD/SD) (also called semantic dementia), (ii) agrammatic PPA (also called progressive non-fluent aphasia) and (iii) logopenic aphasia (lvPPA) (mainly associated with Alzheimer’s disease pathology). Patients with bvFTD show substantially more social cognition deficits than AD patients [12]. Deficits in social cognition, such as severe alternations in social behavior and loss of empathy, can be a diagnostic criterion for FTD patients [13].

Parkinson’s Disease (PD) is a neurodegenerative disorder that has motor and non-motor symptoms. A deficit in social cognition is one non-motor symptom of PD [14]. Deficits in social cognition are associated with problematic social interactions with other people. Therefore, it is easy to deduce that deficits in social cognition have an impact on the patients’ quality of life [14].

The Revised Self-Monitoring Scale (RSMS) was developed by Lennox and Wolfe (1984) and is a 13-item 6-point Likert scale that includes two subcategories [15]: (a) the Expressive Behavior (EX) subscale, which measures the subjects’ sensitivity to the behavior of others, and (b) the Self-Presentation (SP) subscale, which measures the subjects’ tendency to monitor their self-presentation. Both subscales relate to social cognition [16]. This means that high self-monitors are able to understand other people’s expressions in social circumstances as guidelines for managing their own behavior, whereas low self-monitors often guide their behavior from their personal attitudes, beliefs and personality [16]. However, at the same time, there are individuals who understand other people’s behavior, but they are unable to modify their behavior in accordance with the circumstances due to other difficulties, such as personality disorders [16].

The aim of the current study is (a) to examine how the socioemotional parameter is affected in neurovegetative diseases and (b) whether the RSMS is a useful tool in order to discern these disorders based on the socioemotional parameters in the Greek population. Therefore, the hypothesis of the current manuscript is: “the RSMS scale is a useful tool in order to discern the neurodegenerative disorders based on the socioemotional parameter in the Greek population”.

## 2. Materials and Methods

### 2.1. Subjects

A total of 331 subjects (N = 331) participated in the current study. The participants were recruited from the Third Age Centre “IASIS” in Athens (Greece). The study is in accordance with the ethical principles (Declaration of Helsinki). The patients and their caregivers have given written consent in order to participate in the study. Their mean age was 69.1 years (SD = 9.6 years). Except for the control group, participants suffered from Alzheimer’s Disease (AD), behavioral variant Frontotemporal Dementia (bvFTD), Semantic Dementia (svPPA or FTD/SD), Parkinson’s Disease (PD) without dementia and Parkinson’s Dementia (PDD).The criteria for the diagnosis of each disease are in accordance with (a) the National Institute of Neurological and Communicative Disorders and Stroke (NINCDS) and the Alzheimer’s Disease and Related Disorders Association (ADRDA) in order to describe the clinical diagnosis of AD and MCI due to AD [17], (b) the International Behavioral Variant FTD Criteria Consortium (FTDC) in order to describe the diagnosis of bvFTD [18], (c) the FTD/SD classifications [19] and (d) the criteria for Parkinson’s Disease [20]. Hence, 127 (38.4%) of the participants suffered from AD, 47 (14.2%) from bvFTD, 37 (11.2%) from FTD/SD, 21 (6.3%) from PD and 18 (5.4%) from PDD. PD was distinguished from PDD in terms of functionality due to the cognitive deficits that were mentioned in the interview with the patient and the caregiver and the medical diagnosis by experienced neurologists and psychiatrists using MRIs and cognitive tests. PD may also have a pathological social cognition deficit and, for exploratory reasons, the study aimed to examine whether RSMS can discern the social emotion deficits across neurodegenerative diseases. Moreover, another question that arose was whether PD patients without dementia can effectively control their social behavior [21]. The median duration of their disease was 2.0 years (IQR: 0.6–4.0 years). The healthy cohort included 81 participants (24.5%) who did not suffer from any of the aforementioned diseases.

A team of neurologists, psychiatrists and clinical neuropsychologists diagnosed the participants based on the internationally established criteria. The neuropsychological assessments are referred to below. For the patients’ diagnoses, the team also considered their MRIs and history interviews.

### 2.2. Neuropsychological Procedures

The neuropsychological scales have been used in order to measure the cognitive impairments. The MMSE [22] and ACE-R [23] tests are broadly used and they aim to assess the cognitive functions of the participant. In particular, ACE-R includes MMSE (MMSE is a more common and widely used scale and that is why it was also used) and evaluates all the cognitive functions: orientation in time and space, attention and concentration, memory, verbal fluency, language and visual spatial ability. In the current study, the Greek translations of the measurements were used.

Additionally, in order to estimate the behavioral and neuropsychiatric symptoms (BPSD) of the individuals, we also used the Neuropsychiatric Inventory (NPI) [24]. We asked the questions of the NPI to the dementia caregivers in order to record the behavioral symptoms. This scale categorizes the BPSD in 5 clusters: (a) psychosis, (b) depression, (c) apathy, (d) aggressive behavior and (e) agitation. These clusters include twelve behaviors that have been recorded as the most common disorders in dementia patients. These are: delusions, hallucinations, aggressive behavior, depression, anxiety, euphoria, apathy, disinhibition, agitation, aberrant motor behavior, sleep problems and eating disorders. NPI examines the frequency of the behavior based on a Likert scale from 1–4 (1 = rarely happens, 4 = happens almost every day), the severity of the behavior based on a Likert scale from 1–3 (1 = mild distress to the patient, 3 = severe distress to the patient) and also the distress of the caregiver based on a Likert scale from 1–5 (1 = not at all, 5 = extremely). At the end, the psychologist scored the answers by multiplication of the frequency X severity.

### 2.3. Social Cognition Scale, RSMS

Additionally, the neuropsychologist, in collaboration with the caregiver, completed the questions of the RSMS test. There are 13 questions based on a Likert scale from 1 to 6 (where 1 = does not apply at all and 6 = always applies). In the current study, we used the FTLD module 3 edition: https://naccdata.org/data-collection/forms-documentation/ftld-3, (accessed on 8 December 2022). The questions of the scale can be categorized into two categories: (a) RSMS_EX and (b) RSMS_SP.

In the current study, we asked the caregivers of the participants to answer the 13 RSMS questions (both RSMS_EX and RSMS_SP) in order to evaluate the social cognition of the participants. The RSMS (total) is the sum of the two following subscales: (a) the Expressive Behavior (EX) subscale, which measures the individuals’ sensitivity to the expressive behavior of other people (in other words, if an individual has the ability to alter their behavior in accordance with the circumstances in a social situation) and (b) the Self-Presentation (SP) subscale, which measures the individuals’ tendency to observe their self-presentation (in other words, if an individual is able to read other people’s emotions correctly). In the RSMS_EX subscale, items 12, 14, 22 and 23 measure extraversion. RSMS-EX comprises 7 questions, each of which is based on a 6-point Likert scale (always false to always true). The RSMS_SP subscale includes questions 2, 4, 5, 6, 8 and 13 and refers to sensitivity to the expressive behavior of others. RSMS has reliability and construct validity and has been used broadly, also in the Greek population [24].

## 3. Statistical Analyses

Normal distributed variables are expressed as mean (standard deviation), while variables with skewed distribution are expressed as median (interquartile range). Qualitative variables were expressed as absolute and relative frequencies. For the comparison of proportions, chi-square tests were used. Analysis of Variance (ANOVA) and the Kruskal–Wallis test were used for the comparison of continuous variables between all study groups. Analysis of Covariance (ANCOVA) was used for the comparison between RSMS scales among all study groups after adjusting for gender, age, years of education and ACE-R score. ACE-R is a tool for assessing the cognitive deficits and it can be affected by the RSMS. Age and gender were added as typical confounding variables. Bonferroni correction was used in the case of multiple testing in order to control for type I error. The Kolmogorov–Smirnov test was used in order to evaluate the normality of variances. Pearson and Spearman correlation coefficients were used to explore the association between RSMS scales and ACE-R and NPI subscales. Pearson correlation coefficients were used when both variables were normally distributed, while Spearman correlation coefficients were used in the case when at least one variable was not normally distributed. A correlation coefficient between 0.1 and 0.3 was considered low, between 0.31 and 0.5 was considered moderate and those over 0.5 were considered high. Partial correlations of RSMS scales with ACE-R and NPI were also computed in order to investigate whether correlations were significant after adjustment for age, gender, years of education and diagnosis. All reported *p* values are two-tailed. Statistical significance was set at *p* < 0.05 and analyses were conducted using SPSS statistical software (version 22.0). Additional information concerning group comparisons from Table 1 was added in the results section.

## 4. Results

The sample consisted of 331 participants (57.4% females), with mean age 69.1 years (SD = 9.6 years). In addition, 127 (38.4%) of the participants suffered from AD, 47 (14.2%) from bvFTD, 37 (11.2%) from FTD/SD, 21 (6.3%) from PD and 18 (5.4%) from PDD. The median duration of their disease was 2.0 years (IQR: 0.6–4.0 years). The remaining 81 participants (24.5%) did not suffer from any of the aforementioned diseases. Participants’ demographics, clinical characteristics, MMSE, NPI and ACE-R scores are presented in Table 1, in total and by each group. There were significant differences among all study groups regarding participants’ demographics and clinical characteristics. Furthermore, MMSE and ACE-R scores differed significantly among all study groups. All NPI subscales, except for depression, anxiety and irritability, differed significantly among all study groups.

Comparison of participants’ RSMS scores among study groups, after adjusting for gender, age and ACE-R, are presented in Table 2. There were significant differences in RSMS SP (Self-Presentation), RSMS EX (Expressive Behavior) and total RSMS among all study groups. More specifically, after Bonferroni correction, it was found that RSMS SP was significantly higher in the healthy group compared with AD, bvFTD, FTD/SD and PDD. In addition, RSMS SP was significantly lower in the bvFTD group compared with AD and PD. RSMS EX (expressive behavior) was significantly lower in the bvFTD and FTD/SD groups compared with the healthy group and the AD, PD and PDD groups. In addition, RSMS EX and total RSMS were significantly lower in the AD group compared with the healthy group. Total RSMS was significantly lower in the bvFTD group compared with the healthy group and the AD, PD and PDD groups. Moreover, total RSMS was significantly lower in the bvFTD group compared with the healthy group and the AD, PD and PDD groups. Total RSMS was significantly lower in the FTD/SD group compared with the healthy group and the AD and PD groups (Figure 1). Lastly, the RSMS cannot discern AD in nfvPPA patients (Table 3).

Higher values in all ACE-R subscales were associated with greater values in all RSMS scales (Table 4). The aforementioned significant correlations of the RSMS scales with ACE-R and NPI remained significant after adjustment for age, gender, years of education and diagnosis.

According to Table 2, the RSMS total can discern AD in bvFTD (*p* = 0.001) and FTD/SD patients (*p* = 0.005) and AD in the healthy cohort (*p* = 0.001). It is also useful in order to differentiate bvFTD in PD (*p* = 0.002) and PDD patients (*p* = 0.030) and in the healthy cohort (*p* = 0.001). Additionally, it can distinguish FTD/SD in the healthy cohort (*p* = 0.001) and PD patients (*p* = 0.031). On the other hand, the scale cannot discern AD in PD and PDD patients (*p* = 1.000), bvFTD in FTD/SD (*p* = 1.000), FTD/SD in PDD patients (*p* = 0.284), PD patients in the healthy cohort (*p* = 0.195), PDD patients in the healthy cohort (*p* = 0.070) or finally PD in PDD patients (*p* = 1.000).

Specifically, the RSMS_EX subscale can distinguish AD in bvFTD, AD in FTD/SD and AD in the healthy cohort. In addition, it is a useful tool in order to discern bvFTD in the healthy cohort and bvFTD in PD and PDD patients. Additionally, it can differentiate FTD/SD in the healthy cohort and FTD/SD in PD and PDD patients. On other hand, the RSMS_EX subscale cannot discern AD in PD and PDD patients, bvFTD in FTD/SD patients, PD in the healthy cohort or PD in PDD patients.

The RSMS_SP subscale discerns AD in bvFTD patients, AD in the healthy cohort and AD in FTD/SD. Moreover, the subscale distinguishes bvFTD in the healthy cohort, bvFTD in PD and PDD patients and FTD/SD patients in the healthy cohort and PD patients. On the other hand, the RSMS_SP cannot differentiate AD in PD (*p* = 1.000) or PDD (*p* = 1.000) patients. Moreover, the subscale cannot discern bvFTD in FTD/SD patients (*p* = 1.000), FTD/SD in PDD patients, PD in the healthy cohort, PDD patients in the healthy cohort or PD in PDD patients.

Additionally, according to Table 3, the RSMS (total and its two subscales) is weakly correlated with the total score of ACE-R and, more specifically, with the subcategory of ACE-R that refers to verbal fluency (*p* = 0.055). The RSMS (total and the two subscales) is also correlated with the subscales of ACE-R verbal fluency, phonemic and categorical. The RSMS scale (and its two subscales) also has a strong correlation with the NPI inventory. According to the results, there is a correlation between the RSMS total score and its two subscales and subcategory number 7 of the NPI inventory. This subcategory refers to apathy (*p* = 0.050).

## 5. Discussion

This study shows that RSMS (both subscales) can discern the following diseases in the Greek population: (i) AD in bvFTD and in FTD/SD, (ii) AD in the healthy cohort, (iii) bvFTD in the healthy cohort, (iv) bvFTD in PD and (v) FTD/SD in the healthy cohort. Patients with bvFTD showed a statistically significant distinction from AD and PD patients and the healthy cohort with the RSMS scale (and its two subscales). On the other hand, at the same time, our study shows that the RSMS (and both subscales) cannot separate: (i) AD from PD patients, (ii) AD from PDD patients, (iii) bvFTD from FTD/SD patients, (iv) bvFTD from PDD patients, (v) FTD/SD from PD patients, (vi) FTD/SD from PDD patients, (vii) PD from the healthy cohort, (viii) PDD from the healthy cohort and lastly (ix) PD from PDD patients.

Our results are in accordance with previous studies. Toller et al. (2020) also found that the RSMS scale is a valid and sensitive measure for socioemotional changes and that the scale can detect bvFTD. Toller et al. (2018), in an earlier study, also found a relationship between functional SN connectivity and socioemotional sensitivity. This finding is important and in accordance with the current study, because socioemotional sensitivity is a marker of SN functional connectivity and it can identify bvFTD patients, who are characterized by SN changes. The usefulness of the RSMS for the detection of bvFTD patients is also confirmed in this study. RSMS and SN connectivity amongst the bvFTD patients validates the RSMS as a scale that can be useful for the detection of bvFTD patients. Another common ground with our study and Toller et al. (2018) is that the RSMS cannot discern AD in non-fluent variant Primary Progressive Aphasia (nfvPPA) patients. Moreover, another study confirms the correlation between the RSMS and the FTD diagnosis [25]. The study included 730 participants, consisting of 269 healthy controls, and found that the RSMS detects social cognitive impairment in the genetic FTD of C9orf72 gene.

Furthermore, according to our results, the RSMS scale (total) has a correlation with ACE-R total, ACE-R orientation, ACE-R attention, ACE-R memory, ACE-R language, ACE-R verbal fluency (both verbal fluencies: phonemic and categorical) and ACE-R visuospatial ability. Specifically, the RSMS_EX subscale has a correlation with ACE-R total, ACE-R orientation, ACE-R attention, ACE-R memory, ACE-R language, ACE-R verbal fluency (both verbal fluencies: phonemic and categorical) and ACE-R visuospatial ability. The same correlation applies to the RSMS_SP subscale. Orientation is located in three domains of a single region in the inferior parietal lobe [26]. Orientation in places has found activation in other regions other than the inferior parietal lobe, such as precuneus, medial prefrontal cortex, lateral frontal and temporal lobes and posterior cingulate cortices [27]. Hence, orientation is located across domains, possibly related to the default-mode network. The RSMS scale and its two subscales are related to orientation and therefore are correlated with the aforementioned regions in the brain. Moreover, the front of the brain behind the forehead, which is called the frontal lobe, is responsible for attention. The RSMS scale also has a correlation with the frontal lobe. Memory is located in the temporal lobes, especially the hippocampus. The frontal versus the temporal cortex is related to verbal fluency; specifically, categorical fluency is located in the temporal cortex. The verbal fluency of the ACE-R test includes both a phonemic and a semantic test. The frontal lobe is primarily involved in phonemic fluency, whereas semantic fluency tests typically tap on the anterior temporal lobe function. Verbal fluency is located in the frontal and temporal lobes; these lobes are mainly affected in patients with FTD. Hence, the RSMS is also correlated with the frontal and temporal lobes. Finally, the visuospatial ability is placed in the frontal and posterior areas, where saccades are also influences. The posterior parietal cortex is also involved in the visuospatial ability. Our results are in accordance with previous studies. Parthimos et al. have found that the RSMS scale is associated with alterations in the bilateral temporal and frontal lobes, cingulum, putamen, precuneus, olfactory, hippocampus, fusiform, parahippocampal and insula.

In addition, there is a correlation between the RSMS scale and the NPI total and these three domains: apathy, disinhibition and eating disorders. According to a previous study, these behaviors are related to the frontal lobes [28,29,30]. This is a critical finding, because these three abovementioned behaviors are very common in patients with FTD. Therefore, it seems that the RSMS is a scale that is also associated with a cognitive test (ACE-R) and a behavioral questionnaire (NPI). Hence, (a) cognitive deficits in general (particularly in the verbal fluency), (b) in combination with BPSD problems (especially the three abovementioned behaviors) and (c) problems in social cognition in general (particularly in the ability to modify their behavior (RSMS_SP) and the ability to understand the behavior of others (RSMS_EX)) construct the profile of an FTD patient. To our knowledge, no other study has examined the correlation with the RSMS scale and its two subscales with apathy, disinhibition or eating disorders.

Our study has some limitations. First of all, the sample size is not large. Moreover, the RSMS cannot discern FTD subtypes (bvFTD in FTD/SD). Secondly, the PD and PDD participants in our study are few. Thirdly, the scale is based on the caregiver; this means that it may not measure objectively, because the caregiver may overstate or underate the patient’s behavior. Finally, the patient’s personality may be another limitation. There are people who do not have a correct perception of the social environment or they have difficulties in adapting to the new environment even though their perception is normal. Future studies should focus on finding valuable and trustful scales in order to discern the FTD subtypes because they share some common neuroanatomical deficits and have similar BPSD and socioemotional deficits and many times their differential diagnosis is difficult [31]. On the other hand, the strength of our study is that our results are in accordance with previous studies and the results in the Greek population can also be applied to the general population of patients with neurodegenerative diseases.

## 6. Conclusions

In sum, the RSMS is a helpful tool in order to identify socioemotional deficits in neurodegenerative diseases. It is also a useful scale that can separate FTD in AD patients. Furthermore, it is a brief questionnaire that can be used in order to identify the earliest changes in socioemotional behavior in bvFTD patients. The RSMS scores (total and its two subscales) correlate with specific cognitive abilities, such as orientation, attention, memory, language and visuospatial ability, as well as some BPSD. It is a scale that can reliably measure the social behavior (adaptations of the social environment, modification of the behavior and understanding others’ behavior) in non-reversible neurodegenerative diseases in the Greek population, such as AD, FTD, PDD and PD.

## Figures and Tables

**Figure 1 jcm-11-07375-f001:**
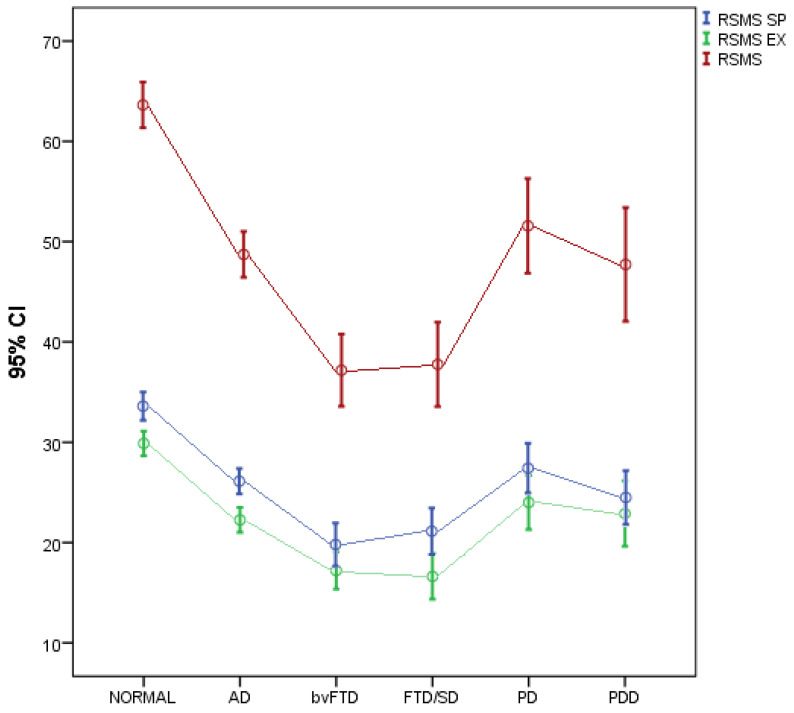
Participants’ RSMS scores for each study group. Note: FTD/SD = svPPA.

**Table 1 jcm-11-07375-t001:** Sample characteristics in total and by group.

	Total Sample N = 331; 100%	Healthy Group ^a^ N = 81; 24.5%	AD ^b^ N = 127; 38.4%	bvFTD ^c^ N = 47; 14.2%	svPPA or FTD/SD ^d^ N = 37; 11.2%	PD ^e^ N = 21; 6.3%	PDD ^f^ N = 18; 5.4%	*p*
Gender, N (%)								
Males	141 (42.6)	27 (33.3)	53 (41.7)	17 (36.2)	22 (59.5)	15 (71.4)	7 (38.9)	0.009 ‡
Females	190 (57.4)	54 (66.7)	74 (58.3)	30 (63.8)	15 (40.5)	6 (28.6)	11 (61.1)	
Age (years), mean (SD)	69.1 (9.6)	62.5 (9.4) ^b,d,e,f^	72.8 (8.4) ^a,c^	65.9 (9.4) ^b,e,f^	68.7 (7.6) ^a,f^	73.6 (6.3) ^a,c^	76.7 (6.9) ^a,c,d^	<0.001 +
Years of education, mean (SD)	11.4 (5.2)	13.7 (2.9) ^b,c,d,f^	10.7 (6.4)^a^	10.5 (4.2) ^a^	10.2 (4.4) ^a^	12.7 (4.1)	9 (4.6) ^a^	<0.001 +
Years of disease, median (IQR)	2 (0.6–4)	-	3 (2–4)	3 (1.6–5)	4 (2–5)	3 (2–6)	3 (2–5)	0.354 ++
MMSE, mean (SD)	23.4 (6.2)	29.4 (0.8) ^b,c,d,f^	22 (4.8) ^a,c,d,e^	19.4 (7) ^a,b,e^	18.8 (7.8) ^a,b,e^	27.2 (1.3) ^b,c,d,f^	21.7 (3.6) ^a,e^	<0.001 +
NPI delusions, median (IQR)	0 (0–0)	0 (0–0)	0 (0–0) ^c,d^	0 (0–3) ^b^	0 (0–2) ^b^	0 (0–0)	0 (0–2)	<0.001 ++
NPI hallucinations, median (IQR)	0 (0–0)	0 (0–0) ^f^	0 (0–0) ^f^	0 (0–0) ^f^	0 (0–0)	0 (0–0)	0 (0–1) ^a,b,c^	<0.001 ++
NPI agitation/aggressive behavior, median (IQR)	0 (0–2)	0 (0–0) ^d^	0 (0–1) ^d^	0 (0–2)	2.5 (0–4) ^a,b^	1 (0–3)	0 (0–4)	<0.001 ++
NPI depression, median (IQR)	0 (0–3)	0 (0–2)	0 (0–3)	0 (0–3)	1.5 (0–3.5)	1 (0–4)	3 (0–4)	0.091 ++
NPI anxiety, median (IQR)	0 (0–3)	0 (0–4)	0 (0–3)	0 (0–3)	1 (0–4)	2 (0–4)	3 (0–6)	0.383 ++
NPI euphoria, median (IQR)	0 (0–0)	0 (0–0) ^c,d^	0 (0–0) ^c^	0 (0–3) ^a,b^	0 (0–1.5) ^a^	0 (0–0)	0 (0–0)	<0.001 ++
NPI apathy, median (IQR)	3 (0–8)	0 (0–0) ^b,c,d,e,f^	3 (0–4) ^a,c^	6 (3–12) ^a,b^	3 (2–8) ^a^	3 (2–8) ^a^	3 (3–6) ^a^	<0.001 ++
NPI disinhibition, median (IQR)	0 (0–1)	0 (0–0) ^c,d^	0 (0–0) ^d^	0 (0–3) ^a^	0.5 (0–4) ^a,b^	0 (0–0)	0 (0–0)	0.001 ++
NPI irritability, median (IQR)	0 (0–3)	0 (0–3)	0 (0–3)	0 (0–2)	2 (0–5)	2 (0–3)	3 (0–6)	0.093 ++
NPI wandering, median (IQR)	0 (0–0)	0 (0–0) ^c,d^	0 (0–0) ^c^	1 (0–8) ^a,b,f^	0 (0–3) ^a^	0 (0–0)	0 (0–0) ^c^	<0.001 ++
NPI sleeping problems, median (IQR)	0 (0–3)	0 (0–0) ^d,e,f^	0 (0–3)	0 (0–4)	1.5 (0–4) ^a^	3 (0–4) ^a^	2 (0–6) ^a^	0.004 ++
NPI eating disorders, median (IQR)	0 (0–4)	0 (0–0) ^c,d,e,f^	0 (0–3) ^c^	4 (0–8) ^a,b^	0 (0–5) ^a^	0 (0–4) ^a^	1 (0–4) ^a^	<0.001 ++
NPI TOTAL, median (IQR)	16 (8–27)	7 (3–12) ^b,c,d,e,f^	13 (6–19) ^a,c,d^	24.5 (15–38) ^a,b^	25.5 (12–43.5) ^a,b^	15.5 (10–28) ^a^	25 (15–31) ^a^	<0.001 ++
ACE R, mean (SD)	68.9 (21.7)	94.4 (3.8)^b,c,d,e,f^	63.4 (15.3) ^a,c,d,e^	54.2 (22.1) ^a,b,e^	51.1 (21.8) ^a,b,e^	78.6 (8) ^a,b,c,d,f^	56.6 (9.2) ^a,e^	<0.001 +
ACE R ORIENT, mean (SD)	8.1 (2.3)	10 (0.2) ^b,c,d,f^	7.5 (2.1) ^a,e^	6.9 (2.4) ^a,e^	6.9 (3.3) ^a,e^	9.7 (0.6)^b,c,d,f^	7.7 (1.7) ^a,e^	<0.001 +
ACE R ATTENT, mean (SD)	6.8 (1.8)	8 (0.1) ^b,c,d,f^	6.4 (1.8) ^a,e^	5.6 (2.2) ^a,e^	6.1 (2.2) ^a,e^	7.8 (0.4) ^b,c,d^	6.4 (1.6) ^a^	<0.001 +
ACE R MEM, mean (SD)	15.5 (7.2)	24.2 (2.2) ^b,c,d,e,f^	12.1 (5) ^a,e^	12.2 (7.1) ^a,e^	11.1 (5.5) ^a,e^	18.4 (4.6) ^a,b,c,d,f^	11.6 (4.6) ^a,e^	<0.001 +
ACE R VERFLUEN, mean (SD)	6.5 (3.8)	11.3 (1.7) ^b,c,d,e,f^	5.3 (2.5) ^a,c,d,e^	3.7 (2.6) ^a,b,e^	3.7 (2.4) ^a,b,e^	7.1 (2.7) ^a,b,c,d,f^	4.3 (2.1) ^a,e^	<0.001 +
ACE_R_VERFLUEN_PHON, mean (SD)	3.4 (2.0)	5.3 (0.1) ^b,c,d,e,f^	3.1 (0.2)^a,c,d^	1,9 (0.2) ^a,b,e^	1.6 (0.2) ^a,b,e^	3.7 (0.5) ^a^	2.7 (0.4) ^a,c,d^	<0.008 ++
ACE_R_VERFLUEN_CAT, mean (SD)	3.4 (2.1)	5.9 (0.1) ^b,c,d,e,f^	2.7 (0.1) ^a,c,d,f^	1.7 (0.2) ^a,b,e^	1.6 (0.3) ^a,b,e^	4.1 (0.6) ^a^	2.3 (0.5) ^a,b,c,d^	<0.004 ++
ACE R LANG, mean (SD)	21.2 (5.5)	25.7 (0.9) ^b,c,d,f^	20.6 (4.9) ^a,d^	18.7 (5.6) ^a,d,e^	14.8 (7.1) ^a,b,c,e^	23.2 (2.1) ^c,d,f^	18.5 (4.4) ^a,f^	<0.001 +
ACE R VS, mean (SD)	12 (3.9)	15.3 (1) ^b,c,d,e,f^	11.6 (3.2) ^a,c,f^	9.8 (3.9) ^a,b^	10.3 (5.1) ^a^	12.1 (2.9) ^a,f^	8.1 (4.2) ^a,b,e^	<0.001 +

+ ANOVA; ++ Kruskal–Wallis test; ‡ Pearson’s chi square test; ^a,b,c,d,e,f^: significant difference after Bonferroni correction.

**Table 2 jcm-11-07375-t002:** Comparison of participants’ RSMS scores among study groups after adjusting for gender, age, years of education and ACE-R.

	RSMS SP	RSMS EX	RSMS
Mean (SD)	Mean (SD)	Mean (SD)	Mean (SD)
Healthy	33.7 (6.4)	29.9 (5.6)	63.6 (10.3)
AD	26.4 (7.2)	22.3 (7.1)	48.7 (13.1)
bvFTD	20.0 (7.3)	17. 2 (6.3)	37.2 (12.3)
FTD/SD	21.1 (6.9)	16.6 (6.8)	37.8 (12.6)
PD	27.6 (5.4)	24.0 (5.9)	51.6 (10.4)
PDD	24.5 (5.4)	22.9 (6.6)	47.7 (11.4)
Pa	< 0.001	<0.001	<0.001
Pb			
AD vs. bvFTD	<0.001	<0.001	<0.001
AD vs. FTD/SD		0.007	0.005
AD vs. HEALTHY	0.002	0.001	<0.001
bvFTD vs. HEALTHY	<0.001	<0.001	<0.001
bvFTD vs. PD	0.011	0.005	0.002
bvFTD vs. PDD		0.018	0.030
FTD/SD vs. HEALTHY	<0.001	<0.001	<0.001
FTD/SD vs. PD		0.019	0.031
FTD/SD vs. PDD		0.050	
HEALTHY vs. PDD	0.019		
AD vs. bvFTD	<0.001	<0.001	<0.001
AD vs. FTD/SD		0.007	0.005

Note: only significant differences are shown.

**Table 3 jcm-11-07375-t003:** Correlation of RSMS dimensions with ACE-R scales.

	RSMS SP	RSMS EX	RSMS
Mean (SD)	Mean (SD)	Mean (SD)	Mean (SD)
ACE R	0.48a ***	0.48a ***	0.51a ***
ACE R ORIENT	0.34a ***	0.32a ***	0.35a ***
ACE R ATTENT	0.32a ***	0.31a ***	0.34a ***
ACE R MEM	0.43a ***	0.42a ***	0.45a ***
ACE R VERFLUEN	0.52a ***	0.51a ***	0.55a ***
ACE-R VERFLUEN_PHON	0.49b ***	0.50b ***	0.53b ***
ACE-R VERFUEN_CATEG	0.52b ***	0.54b ***	0.57b ***
ACE-R LANG	0.41a ***	0.43a ***	0.45a ***
ACE R VS	0.40a ***	0.37a ***	0.41a ***

a Pearson’s correlation coefficient; b Spearman’s correlation coefficient; *** *p* < 0.001.

**Table 4 jcm-11-07375-t004:** Correlation of RSMS dimensions with NPI scales.

	RSMS SP	RSMS EX	RSMS
NPI delusions		−0.14b *	−0.10b
NPI hallucinations		−0.08b	0.04b
NPI aggressive beh/agitation		−0.28b ***	−0.20b **
NPI depression		−0.08b	−0.06b
NPI anxiety		−0.04b	−0.05b
NPI euphoria		−0.26b ***	−0.20b **
NPI apathy		−0.49b ***	−0.47b ***
NPI disinhibition		−0.38b ***	−0.33b ***
NPI irritability		−0.11b	

b Spearman’s correlation coefficient; * *p* < 0.05; ** *p* < 0.01; *** *p* < 0.001.

## Data Availability

Data are openly available in a public repository.

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
