# Peer review of "The Assessment of the Socioemotional Disorder in Neurodegenerative Diseases with the Revised Self-Monitoring Scale (RSMS)"

_jcm, 2022, doi:10.3390/jcm11247375_

Round 1

Reviewer 1 Report

The manuscript "The assessment of the socioemotional disorder in neurodegenerative diseases with the Revised Self-Monitoring Scale" by Dimitriou and colleagues presents a study in healthy controls and patients with neurodegenerative diseases (namely: Parkinson's, Alzheimer's, frontotemporal dementia, semantic dementia) in a Greek population that underwent neurocognitive testing including the Revised Self-Monitoring Scale (RSMS), pointing out specific differences in scores of this test and its subscales depending on the underlying disease.

In general, the manuscript needs heavy copy-editing and needs to be revised by a native-speaking professional. Many sentences are odd in structure and some words not fitting. At times it is difficult to follow the train of thought. And: Please spell out any abbreviation at first use, regardless of the abbreviations list at the end of the manuscript (RSMS is abbreviated bevore first use! nfvPPA is not spelled out at all)!

The abstract needs to be completely overhauled. The methods mention "331 dementia subjects" which apparently also included healthy controls! Furthermore, it needs cross-checking by a native-speaker.

The introduction is mostly well-written and easily understandable, even though it suffers from the above-mentioned linguistic difficulties. Sadly, this also applies to the research question in the last paragraph.

The methods are adequate and described in sufficient detail.

The results are succinctly presented and understandable from a data perspective; obviously, copy-editing will make the presentation of the results even clearer. The significant differences among all study groups regarding participants' demographics and clinical characteristics are pointed out and accounted for (as much as possible) by performing ANCOVAs of the RSMS scores and its subscales, all of which yielded highly significant results. In the post-hoc analyses the authors compared healthy controls and patients with each of the aforementioned neurodegenerative diseases. Luckily, there is a clear distinction between healthy controls and patients with neurodegenerative disease (all p < 0.001). Furthemore, the data show significant differences between the scores of patients with Alzheimer's versus frontotemporal dementia as well as semantic dementia, frontotemporal dementia vs. Parkinson's and semantic dementia vs. Parkinson's. Additionally, they identify an inverse correlation between RSMS scores and Neuropsychiatric Inventory (NPI) scores. These are very interesting and potentially useful findings. Sadly, the results are very cumbersome to read as well and could profit from a thorough revision.

The discussion is appropriate and carefully weighs the own data in the context of the existing literature, presenting an overall balanced discussion. I cannot verify the following sentence though: "Another 296 common ground with our study and Toller et al (2018) is that the RSMS cannot distinct 297 AD to nfvPPA patients." Where is that stated in the results? Otherwise, limitations are properly addressed.

The abbreviations list is incomplete (nvfPPA) and on the other hand mentions abbreviations not used in the manuscript (MAPT). Please revise!

The JCM template was used, but some sections were removed such as the Data Availability Statement or the Informed Consent Statement. These should be added. The references are not formatted according to the journal guidelines (let alone formatted consistently). This should be addressed in the revision stage.

Author Response

Thank you for your comments. We have revised the abstract and added at the end of the text the data availability and informed consent statement, as well. We have made the corrections in the results methods, and we have strongly tried to improve our language.

The revised lines are in yellow

Reviewer 2 Report

An interesting and well conduced and written report. I suggest that the abbreviations should be defined at the first time they appear in the abstract or in the text. In addition, a brief description of the strengths of the study should be done.

Author Response

Thank you for you comments.

Our changes have been underlined with yellow in the revised text

Reviewer 3 Report

Thanks for the opportunity to review this interesting paper. The aim of the current 36 study is to examine how the socioemotional parameter is affected in the neurodegenerative diseases  and if the Revised-monitoring scale (RSMS) can discern these disorders based on the socioemotional parameter in the Greek population. The RSMS and its two subscales (RSMS_EX and RSMS_SP) can distinct neurodegenerative diseases effectively. The RSMS can distinct bvFTD to AD, AD to healthy cohort, bvFTD to healthy cohort, bvFTD to PD and  FTD/SD to healthy cohort, and concluded it is a useful tool in order to detect frontotemporal dementias. 

I have major concern on the study design, as only Pearson and Spearman correlation coefficients were used to explore the association of two continuous variables. How the coufounding effects being controlled in this case? This directly impact the validity of the study findings.

Author Response

Thank you for your comments.

Our corrections are underlined with yellow in the revised text.

Partial  correlations of RSMS scales with ACE-R and NPI were also calculated, controlling for age, gender, years of education and diagnosis and the results remained significant. The relevant information was added in the results section.

Round 2

Reviewer 3 Report

No further comments. Thanks